# An updated comparison of standard and novel FEV$_1$ indices' association with all-cause mortality

Stephen T. Russell [1,2]*, Mohleen Kang[1,2], Jordan A. Kempker[1]

**1** Division of Pulmonary, Allergy, Critical Care and Sleep Medicine, Department of Medicine, Emory University School of Medicine, Atlanta, Georgia, **2** Atlanta Veterans Affairs Health Care System, Decatur, Georgia

* struss2@emory.edu

## Abstract

### Rationale

Current recommendations for defining FEV$_1$ abnormalities are based on Z-score cutoffs. Alternative approaches may better correlate with patient-related outcomes, including mortality.

### Objective

This study evaluates the association between FEV$_1$ value and mortality in six FEV$_1$ indices in a large, U.S based cohort.

### Methods

This is a cohort of 2007–2012 National Health and Nutrition Examination Study (NHANES) participants with spirometry and linked mortality data through 2019. We transformed FEV$_1$ values to the following indices: raw FEV$_1$, FEV$_1$-Z scores (FEV$_1$-Z), FEV$_1$-Percent Predicted (FEV$_1$-PP), FEV$_1$/Height$^2$ (FEV$_1$/Ht$^2$), FEV$_1$/Height$^3$ (FEV$_1$/Ht$^3$), and FEV$_1$-Q. We compared association with all-cause mortality between lowest and highest FEV$_1$ deciles of each index using Cox Proportional Hazards models. Two sensitivity analyses were performed, one after applying NHANES sample weighting and one including lower quality spirometry. A subgroup analysis of participants with airflow obstruction, defined as FEV$_1$/FVC ≤ LLN, was performed.

### Results

Of the 12,994 included participants, 971 (7.5%) had died. The majority (56.2%) were nonsmokers and 13.2% had an FEV$_1$/FVC < LLN. When comparing the most to least severe deciles of FEV$_1$, the indices with the largest magnitude unadjusted hazard ratios were FEV$_1$/ht$^3$ (HR 26.4, 95%CI 16.0–43.6), FEV$_1$/ht$^2$ (HR 21.8, 95%CI 13.8–34.7), FEV$_1$-Q (HR 17.5, 95%CI 11.6–26.5), and FEV$_1$ (HR 14.4, 95%CI 9.7–21.5). After adjusting for age, gender, and tobacco pack-years, FEV$_1$/Ht$^3$ (HR 4.9, 95%CI

**Data availability statement:** All data used in this study, including demographic information, spirometry measurements, and linked mortality outcomes, are publicly available through the National Health and Nutrition Examination Survey (NHANES) website (https://wwwn.cdc.gov/nchs/nhanes/). Specific datasets include the 2011–2012 NHANES spirometry data (https://wwwn.cdc.gov/nchs/nhanes/search/datapage.aspx?Component=Examination&Cycle=2011-2012), the 2009–2010 spirometry data (https://wwwn.cdc.gov/nchs/nhanes/search/datapage.aspx?Component=Examination&Cycle=2009-2010), and the 2007–2008 spirometry data (https://wwwn.cdc.gov/nchs/nhanes/search/datapage.aspx?Component=Examination&Cycle=2007-2008). Mortality outcomes linked to the National Death Index (NDI) are available through the NHANES Linked Mortality Files (https://www.cdc.gov/nchs/data-linkage/mortality-public.htm).

**Funding:** STR received funding from the National Heart, Lung, and Blood Institute (NHLBI) T32 grant 5T32HL116271. MK received funding from the Department of Veterans Affairs VISN 7 Research Development Award. STR was also supported by the National Center for Advancing Translational Sciences (NCATS) of the National Institutes of Health under Award Number UL1TR002378. The content is solely the responsibility of the authors and does not necessarily represent the official views of the National Institutes of Health.

**Competing interests:** The authors have declared that no competing interests exist.

2.6–9.3) and $FEV_1/Ht^2$ (HR 4.8, 95%CI 2.7–8.7) had the highest hazard ratios, however the confidence intervals had significant overlap with other indices. In adjusted analyses, the C-statistic (0.81) was the same across indices. Sensitivity and subgroup analyses yielded a similar pattern.

## Conclusions

Reference-range-independent indices based on absolute $FEV_1$ (raw $FEV_1$, $FEV_1$-Q, $FEV_1/Ht^2$, $FEV_1/Ht^3$) are equally or more strongly associated with mortality than reference-range- dependent $FEV_1$ indices ($FEV_1$-Z scores, $FEV_1$-Percent Predicted) in a large U.S. cohort.

## Introduction

Spirometry has long been used for diagnosing and monitoring respiratory diseases, however, the approach to define respiratory impairment in a patient or a population remains controversial. The 2022 European Respiratory Society and American Thoracic Societies' *(ERS/ATS) Technical Standard On Interpretive Strategies For Routine Lung Function Tests* newly recommends using Z-scores of forced expiratory volume in 1 second ($FEV_1$-Z) in place of percent predicted ($FEV_1$-PP) values to define abnormality and severity [1]. More recently, the ATS recommended against utilizing race- and ethnicity-specific coefficients in reference equations [2]. Work in the past couple of decades has demonstrated the potential of alternative approaches of indexing $FEV_1$ with the development of the $FEV_1$ quotient ($FEV_1$-Q), which standardizes the $FEV_1$ to the 1st percentile of predicted $FEV_1$ from population-based samples; and examining $FEV_1$ divided by exponents of standing height ($FEV_1/ht^2$ and $FEV_1/ht^3$) [3–11]. Miller and Pedersen developed $FEV_1$-Q and showed that $FEV_1$-Q, $FEV_1/ht^2$, and $FEV_1/ht^3$ outperformed $FEV_1$-PP and $FEV_1$-Z in predicting mortality [5]. Additionally, $FEV_1$-Q and $FEV_1/ht^3$ were demonstrated to be the best predictors of mortality in an Italian chronic obstructive pulmonary disease (COPD) cohort [6]. Based on these findings the ERS/ATS guidelines reference $FEV_1$-Q as a promising approach to both defining lung impairment and tracking changes in individual patients over time, but recommend further investigation [1].

Reference-range-independent spirometric indices have the potential to avoid some of the biases introduced by using population-based reference equations to standardize a raw $FEV_1$ value. Development and maintenance of reference ranges for spirometry rely on large samples of individuals which are costly and carry the potential of bias when applied to other, external populations [2]. Reference-range-independent indices such as $FEV_1$-Q, $FEV_1/ht^2$ and $FEV_1/ht^3$ are less affected by reference range induced bias and may be utilized to define lung impairment and its severity. They have shown promise for their strong associations to clinical outcomes but require further validation and exploration.

Within this framework, our study compares the associations of both reference-range-dependent ($FEV_1$-PP, $FEV_1$-Z) and reference-range-independent ($FEV_1$, $FEV_1$-Q, $FEV_1/ht^2$ and $FEV_1/ht^3$) indices with all-cause mortality.

## Methods

### Data source

The NHANES is a nationwide, complex, multi-stage sample of non-institutionalized individuals living in the US, now conducted in two-year cycles by personnel from the National Center for Health Statistics (NCHS). We combined and analyzed data from three NHANES cycles (2007–2008, 2009–2010, and 2011–2012) as these years included spirometry measurements during the physical examination portion of the survey. For consenting NHANES participants, the NCHS additionally provides linkage of their survey data to the National Death Index (NDI) and provides linked public-use data files which we utilized to build a cohort study for examining the associations between baseline $FEV_1$ indices and all-cause mortality through 2019. The public-use files only provide data on adult survey participants and NCHS performs selected data perturbation techniques to reduce risk of participant identification [12]. The three cycles of NHANES included in our analyses were approved by the NCHS Research Ethics Review Board. Specifically, the 2007–2008, 2009–2010, and 2011–2012 NHANES cycles were approved under Continuation of Protocol #2005–06, Continuation of Protocol #2005–06, and Protocol #2011–17, respectively. Participants provided written consent to participate in NHANES. Our analyses were conducted on fully de-identified, publicly available data and therefore did not require additional IRB approval. Data were accessed and downloaded 7/25/2023.

### Spirometry

The NHANES performed spirometry on individuals from 6–79 years old, although only included those ≥ 18 years-old in the public-use dataset. Their protocol excluded the following participants from spirometry, those: with current chest pain; with a physical problem with forceful expiration; on supplemental oxygen; with recent surgery of the eye, chest, or abdomen; with a recent heart attack or stroke; with a personal history of detached retina or pneumothorax; or with a tuberculosis exposure or had recently coughed up blood. NHANES used an Ohio Sensormed 827 dry rolling seal spirometer through all the selected survey years and testing was completed according to the ATS standards for acceptability and reproducibility at the time of study enrollment [13]. Each spirometry test was graded on a scale from A-F according to ATS standards [14]. Participants meeting grades A and B were included in our primary analysis, and those meeting grades A, B, C, or D were included in a sensitivity analysis.

### $FEV_1$ indices

The percent predicted $FEV_1$ ($FEV_1$-PP) represents the ratio of an individual's raw $FEV_1$ value in relation to the mean $FEV_1$ from a reference population of healthy individuals [1,15]. $FEV_1$-Z captures the number of standard normal deviations an individual's raw $FEV_1$ value lies from the mean of a normalized distribution of $FEV_1$. For both $FEV_1$-PP and $FEV_1$-Z calculation, we utilized the Global Lung Function Initiative (GLI) reference equations accounting for age, sex, height, and ancestry. For ancestry we utilized the "GLI-Other" for all participants in accordance with the 2023 update from ATS/ERS which recommended adopting race-neutral reference ranges for spirometry [2]. In this study we utilize the original gender-specific Q values for men and women (0.5L for men and 0.4L for women) to derive $FEV_1$-Q. The $FEV_1$/height$^n$ indices are calculated as the raw $FEV_1$ value divided by the square or cubic exponent of standing height in centimeters. The raw $FEV_1$ index is a pre-bronchodilator value.

### Statistical analysis

For our analyses estimating the associations between each $FEV_1$ index with all-cause mortality, we first categorized each $FEV_1$ index into deciles of worsening $FEV_1$ severity. Categorizing each index into deciles is a methodology used in previous papers to compare indices that may have different or unknown severity cutpoints [5]. We then performed Cox Proportional Hazards regression analysis for each $FEV_1$ index decile group, using the least severe decile group as the

referent. For each $FEV_1$ index we performed both an unadjusted regression and a regression adjusted for gender, age, and tobacco pack-years. We compared index performance by the magnitude of the hazard ratio of the most severe $FEV_1$ index decile group and the model's overall c-statistic.

All data management, the estimate of $FEV_1$ first percentiles, and sensitivity analyses using survey weights were performed using SAS software [Version 9.4. Copyright © SAS Institute Inc.] and regression analyses were performed using RStudio [Version 2023.06.1, © 2009–2023 Posit Software, PBC] with the *survival* software package [Version 3.5–7 © 2000 Mayo Foundation for Medical Education and Research].

### Subgroup and sensitivity analyses

Since the NHANES cohort is a community dwelling cohort, which may have a high proportion of individuals without lung disease, we also performed a subgroup analysis on those participants with airflow limitation, defined by $FEV_1$/Forced Vital Capacity (FVC) less than the expected lower limit of normal (LLN).

We performed two sensitivity analyses. First, we repeated the primary analyses after applying NHANES survey weights, which approximate nation-wide representation, to the sample using the SAS *survey* procedures. Second, we included subjects with spirometry ATS grades A, B, C, and D. Our primary analysis included only ATS quality grade A and B to increase internal validity. ATS guidelines suggest that grades C and D studies may produce usable and acceptable maneuvers but should be interpreted with caution, and that grade F studies are unusable. We therefore performed a sensitivity analysis including ATS grades C and D but excluding grade F. For subgroup and sensitivity analyses the $FEV_1$ index deciles were re-calculated and Cox Proportional Hazards regression was repeated as per the primary analysis. These SAS procedures do not produce a c-statistic and we were therefore unable to include this value in the sensitivity analyses.

## Results

The three NHANES cohorts from 2007 and 2012 included 30,442 total participants with 16,411 aged 18–79 years participating in the exam portion of the survey and with their data linked to the NDI. After additional exclusions applied by the NHANES personnel for eligibility for spirometry and our own exclusions for poor quality spirometry, there were 12,994 participants (50.4% women) included in the primary analysis (Fig 1). At the time of spirometry, the study sample had a median age of 44 years (interquartile range (IQR) 31–59 years), median height of 168 cm (IQR 160–175 cm), and median $FEV_1$ of 3.0 L (IQR 2.5-3.7L). The majority (56.2%) reported no smoking history, with 8.7% reporting >20 pack-years of tobacco exposure and 13.2% with an $FEV_1$/FVC ≤ LLN. By the end of 2019, 971 (7.5%) of the participants had died (see Supplementary Table E1 for more detailed descriptive statistics).

### $FEV_1$ indices' associations with all-cause mortality

The ranges of raw $FEV_1$ values amongst decile groups within each $FEV_1$ index are shown in Supplementary Table E2. Table 1 shows the results of the unadjusted regression analyses for all-cause mortality across deciles of each $FEV_1$ index. When compared to the least severe decile, the $FEV_1$ index with the largest magnitude hazard ratio was $FEV_1/ht^3$ (HR 26.4, 95%CI 16.0–43.6), followed by $FEV_1/ht^2$ (HR 21.8, 95%CI 13.8–34.7), $FEV_1$-Q (HR 17.5, 95%CI 11.6–26.5), $FEV_1$ (HR 14.4, 95%CI 9.7–21.5;), $FEV_1$-Z (HR 5.6, 95%CI 4.1–7.6), and $FEV_1$-PP (HR 4.6, 95%CI 3.5–6.0).

Table 2 demonstrates the results of the regression analyses for all-cause mortality across deciles of each $FEV_1$ index when adjusted for age, gender, and tobacco-use history. The c-statistic was the same (0.81) across all the adjusted models. The $FEV_1$ indices with the largest magnitude adjusted hazard ratios were $FEV_1/ht^3$ (HR 4.9, 95%CI: 2.6–9.3) and $FEV_1/ht^2$ (HR 4.8, 95%CI 2.7–8.7). The $FEV_1$ index with the smallest magnitude adjusted HR was $FEV_1$-Q (HR 2.7, 95%CI 1.6–4.4); however, there was overlap in confidence intervals with the other indices.

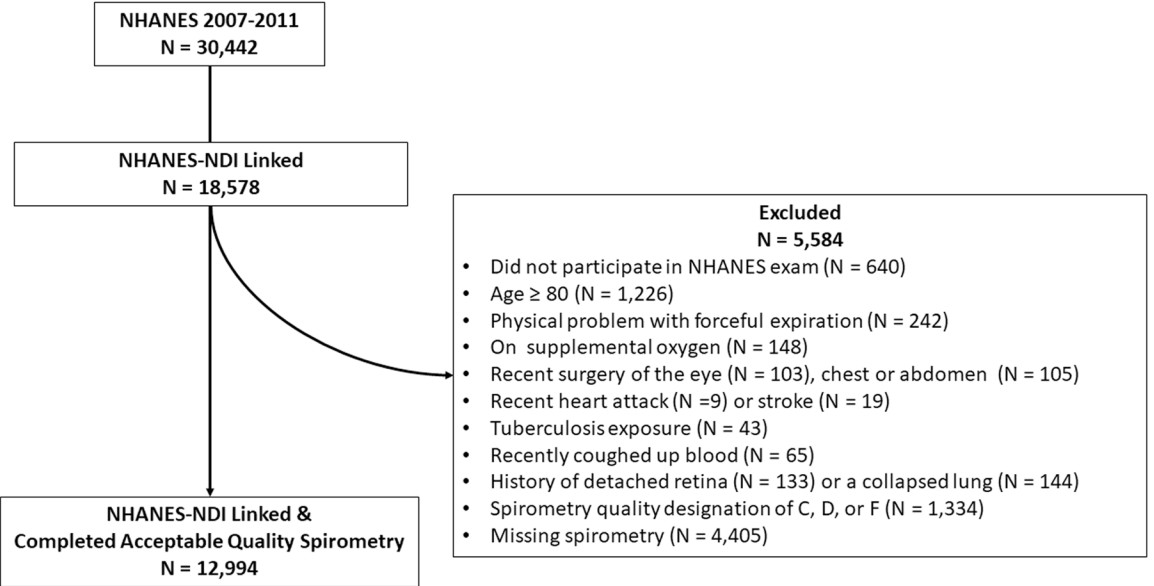

**Fig 1. Cohort Flow Diagram, NHANES subjects with NDI Linkage and Spirometry Exam 2007-2011.** This figure shows the initial NHANES cohort of 30,442 and the exclusions to reach the analyzed sample of 12,994.

**Table 1. Unadjusted cox proportional hazards models for mortality by FEV1 index deciles.**

| Index | $FEV_1$ | | $FEV_1$-Q | | $FEV_1$-Z Scores | | $FEV_1$-Percent Predicted | | $FEV_1$/height$^2$ | | $FEV_1$/height$^3$ | |
|---|---|---|---|---|---|---|---|---|---|---|---|---|
| Decile | HR | 95% CI | HR | 95% CI | HR | 95% CI | HR | 95% CI | HR | 95% CI | HR | 95% CI |
| 10* | — | — | — | — | — | — | — | — | — | — | — | — |
| 9 | 1.1 | 0.7-1.9 | 1.0 | 0.5-1.7 | 1.3 | 0.9-1.9 | 0.9 | 0.6-1.3 | 1.0 | 0.5-1.9 | 1.2 | 0.6-2.3 |
| 8 | 1.8 | 1.1-2.9 | 0.8 | 0.5-1.5 | 1.4 | 1.0-2.0 | 1.0 | 0.7-1.4 | 1.9 | 1.1-3.3 | 1.7 | 0.9-3.2 |
| 7 | 2.4 | 1.5-3.9 | 1.6 | 1.0-2.7 | 1.5 | 1.1-2.2 | 0.9 | 0.6-1.3 | 1.8 | 1.0-3.1 | 2.8 | 1.6-5.0 |
| 6 | 3.0 | 1.9-4.6 | 1.6 | 1.0-2.7 | 1.4 | 1.0-2.0 | 0.8 | 0.6-1.2 | 2.8 | 1.6-4.7 | 2.2 | 1.2-3.9 |
| 5 | 2.6 | 1.6-4.0 | 3.5 | 2.2-5.5 | 1.6 | 1.1-2.2 | 0.9 | 0.7-1.3 | 4.0 | 2.4-6.6 | 3.9 | 2.3-6.8 |
| 4 | 4.1 | 2.6-6.2 | 3.9 | 2.5-6.1 | 2.4 | 1.7-3.3 | 1.5 | 1.1-2.0 | 5.1 | 3.1-8.3 | 7.1 | 4.2-12.0 |
| 3 | 3.7 | 2.4-5.7 | 4.9 | 3.1-7.6 | 2.2 | 1.6-3.1 | 1.3 | 1.0-1.8 | 6.4 | 3.9-10.4 | 8.6 | 5.1-14.5 |
| 2 | 6.4 | 4.2-9.7 | 8.8 | 5.8-13.4 | 3.2 | 2.3-4.4 | 2.1 | 1.6-2.8 | 9.7 | 6.0-15.6 | 11.6 | 7.0-19.4 |
| 1 | 14.4 | 9.7-21.5 | 17.5 | 11.6-26.5 | 5.6 | 4.1-7.6 | 4.6 | 3.5-6.0 | 21.8 | 13.8-34.7 | 26.4 | 16.0-43.6 |
| $R^2$ | 0.05 | | 0.07 | | 0.02 | | 0.03 | | 0.07 | | 0.07 | |
| C (SE) | 0.71 (0.01) | | 0.76 (0.01) | | 0.65 (0.01) | | 0.66 (0.01) | | 0.75 (0.01) | | 0.76 (0.01) | |

*Least severe decile

Abbreviations: $FEV_1$: forced expiratory volume in 1 second, $FEV_1$-Q: $FEV_1$ quotient, HR: hazard ratio, CI: confidence interval

## Subgroup and sensitivity analyses

We performed one subgroup analysis and two sensitivity analyses. For the subgroup analysis of participants with airflow limitation, the hazard ratios for $FEV_1$ indices' association with all-cause mortality had large and overlapping confidence intervals in both unadjusted and adjusted analyses. For the sensitivity analysis use NHANES survey weights to approximate a nationwide sample, $FEV_1$/ht$^3$ and $FEV_1$/ht$^2$ had the highest hazards ratios, however confidence intervals were wide

**Table 2. Adjusted^ cox proportional hazards models for mortality by $FEV_1$ index deciles.**

| Index | $FEV_1$ | | $FEV_1$-Q | | $FEV_1$-Z Scores | | $FEV_1$-Percent Predicted | | $FEV_1$/height$^2$ | | $FEV_1$/height$^3$ | |
|---|---|---|---|---|---|---|---|---|---|---|---|---|
| Decile | HR | 95% CI | HR | 95% CI | HR | 95% CI | HR | 95% CI | HR | 95% CI | HR | 95% CI |
| 10* | — | — | — | — | — | — | — | — | — | — | — | — |
| 9 | 0.8 | 0.4-1.4 | 0.9 | 0.5-1.7 | 1.1 | 0.8-1.7 | 1.1 | 0.8-1.5 | 1.0 | 0.5-2.0 | 1.2 | 0.6-2.5 |
| 8 | 1.1 | 0.7-1.8 | 0.6 | 0.3-1.1 | 1.3 | 0.9-1.8 | 1.3 | 0.9-1.8 | 1.5 | 0.8-2.8 | 1.5 | 0.7-3.0 |
| 7 | 1.3 | 0.8-2.1 | 1.0 | 0.6-1.7 | 1.4 | 1.0-2.0 | 1.3 | 0.9-1.8 | 1.2 | 0.7-2.3 | 1.9 | 1.0-3.8 |
| 6 | 1.6 | 1.0-2.6 | 0.8 | 0.4-1.3 | 1.3 | 0.9-1.8 | 1.2 | 0.8-1.7 | 1.7 | 0.9-2.0 | 1.2 | 0.6-2.4 |
| 5 | 1.2 | 0.7-2.0 | 1.4 | 0.8-2.3 | 1.3 | 0.9-1.9 | 1.2 | 0.9-1.7 | 1.9 | 1.0-3.4 | 1.7 | 0.9-3.3 |
| 4 | 1.8 | 1.1-2.9 | 1.3 | 0.8-2.2 | 2.1 | 1.5-2.9 | 1.9 | 1.4-2.6 | 2.1 | 1.1-3.8 | 2.5 | 1.3-4.8 |
| 3 | 1.5 | 0.9-2.4 | 1.2 | 0.7-2.0 | 1.7 | 1.2-2.4 | 1.6 | 1.1-2.1 | 2.1 | 1.2-3.8 | 2.5 | 1.3-4.8 |
| 2 | 2.1 | 1.3-3.4 | 1.8 | 1.1-3.0 | 2.3 | 1.6-3.2 | 2.1 | 1.6-2.9 | 2.8 | 1.5-5.0 | 2.9 | 1.5-5.5 |
| 1 | 3.8 | 2.4-6.2 | 2.7 | 1.6-4.4 | 3.5 | 2.6-4.9 | 3.1 | 2.3-4.0 | 4.8 | 2.7-8.7 | 4.9 | 2.6-9.3 |
| $R^2$ | 0.11 | | 0.11 | | 0.11 | | 0.11 | | 0.11 | | 0.11 | |
| C | 0.81 (0.01) | | 0.81 (0.01) | | 0.81 (0.01) | | 0.81 (0.01) | | 0.81 (0.01) | | 0.81 (0.01) | |

^Model adjusted for age, tobacco use history, and gender

*Least severe decile

Abbreviations: $FEV_1$: forced expiratory volume in 1 second, FVC: forced vital capacity, $FEV_1$-Q: $FEV_1$ quotient, HR: hazard ratio, CI: confidence interval

and overlapping with other $FEV_1$ indices (see Supplement Tables E5 and E6 for full results). A second sensitivity analysis included 1,163 additional participants with lower quality spirometry grades C and D. This analysis of the 14,157 participants with ATS grades A, B, C, and D showed similar results to the primary analysis (see Supplement Tables E7 and E8 for full results).

## Discussion

In this study of a large sample of US adults, there are several notable findings. When comparing the associations between $FEV_1$ indices and all-cause mortality, results were different based on regression model composition and clinical subgroup and failed to identify a consistently superior $FEV_1$ index. Specifically, in the unadjusted analyses of the entire study population, the results demonstrated that the reference-range-independent indices ($FEV_1$/ht$^3$, FEV1/ht$^2$, $FEV_1$-Q, and $FEV_1$) had stronger associations with all-cause mortality than reference-range-dependent indices ($FEV_1$-Z and $FEV_1$-PP). However, in models further adjusted for age, sex, and tobacco use history and in the subgroup analyses of participants with obstruction, the results are not conclusive with all confidence intervals overlapping.

Before comparing our study with the existing literature, we contextualize our results within our study's strengths and limitations. The major strength of this work rests on the rigorous methodology of the NHANES study sample and its survey and physical examination protocols. The NHANES is a complex, multi-stage national sample that is designed to be representative of community-dwelling US population. Generalizability of our study sample is further strengthened by combining 3 cycles of NHANES, each representing a unique, generalizable sample of US adults. Our sensitivity analysis, applying NHANES survey weights to approximate nationwide representation, enhances the robustness of our analyses. All spirometry was performed by trained NCHS personnel, following ATS technical standards, and utilizing the same equipment across survey years. Additionally, we only included those with the highest quality spirometry results (ATS categories A and B) to strengthen the reliability of our primary analysis. Finally, the NDI is a complete index of death certificates in the US and therefore strengthens the reliability of follow-up for our outcome of all-cause mortality.

A principal limitation of this study is the sole reliance on comparisons of $FEV_1$ indices with all-cause mortality. While this is a highly relevant outcome, other pulmonary health outcomes such as respiratory exacerbations, respiratory-related

hospitalizations, and symptoms are also important in discerning the performance of $FEV_1$ indices in grading the severity of pulmonary morbidity. Additionally, while not a limitation to our study's internal validity, the criteria the NHANES study applies to exclude its participants from conducting spirometry provides important context for interpreting and generalizing our results. Their exclusions of those with recent acute cardiovascular and cerebrovascular conditions may improve the robustness of our results since such conditions may produce impaired spirometry testing related to those underlying conditions rather than intrinsic abnormalities of lung function. The NHANES exclusion from spirometry of those ≥80 years-old or on supplemental oxygen are relevant as they affect this study's findings and generalizability to the elderly and those with chronic hypoxemic respiratory failure, specific target populations for which the estimation of the association between an $FEV_1$ index and mortality may be applied for such clinical decisions as thoracic surgery.

Our analyses raise some relevant questions to the larger goal of comparing $FEV_1$ indices, that while not unique to our specific study, are worth detailing given that they affect the interpretation of our and others' results. The first is whether the adjusted or unadjusted regression represented the more valid approach to comparisons. Some of the indices, particularly the reference-range-dependent indices, already have components of age and/or gender in their composition, making additional adjustments for these covariates in regression modeling seemingly redundant. This, along with the practical consideration that clinically we would like an $FEV_1$ index to inform us about the patient in front of us without having to additionally, conceptually adjust for other factors, seems to favor the comparisons based on unadjusted analyses. On the other hand, not examining analyses adjusted for factors known to be associated with mortality and $FEV_1$ seems to allow a potential for confounding. While we do not propose an answer as to whether adjusted or unadjusted analyses are the more internally valid method for our comparisons, it is for these reasons we decided to present both and globally examine whether results and conclusions differed.

The second methodological dilemma our results raise is whether the results from a broader, more generalized population or a more focused population defined by airflow limitation carry more weight in the comparisons between $FEV_1$ indices. Our underlying conceptual framework at the outset of the study was that each analysis provides information that may be more applicable to a specific context of how the index would be utilized. Analyses of general populations may speak more to an index's capacity for disease case finding while analyses in more specific populations with respiratory impairment may be more applicable to an index's capacity to grade severity and guide clinical decision-making. It is within this context that we interpret our results.

Our comparisons of six $FEV_1$ indices are based on associations with all-cause mortality in survival analyses across both unadjusted and adjusted analyses in both a larger population and a subgroup with airflow limitation. Perhaps it is not surprising that there did not emerge one clear index that was superior to all others. Despite the complexity of analyses and results, we think there are some focused observations that may be informative. In the larger population, applying the criterion of the largest hazard ratio amongst the most severe decile, indices primarily dependent on reference ranges, $FEV_1$-Z and $FEV_1$-PP, performed worse than $FEV_1$, $FEV_1Q$, $FEV_1/height^2$, and $FEV_1/height^3$. In unadjusted analyses the hazard ratio confidence intervals for $FEV_1$, $FEV_1Q$, $FEV_1/height^2$, and $FEV_1/height^3$ were overlapping, but the c-statistic was higher for $FEV_1Q$, $FEV_1/height^2$, and $FEV_1/height^3$ than for raw $FEV_1$. However, in adjusted analyses, the confidence intervals all overlap, and c-statistics were similar. In subgroup analysis of subjects with airflow limitation results were similar, however caution is warranted due to very wide confidence intervals. Other studies have conducted similar analyses, and it is important to put these results in context with other studies.

A comparison of the $FEV_1$ indices association with mortality has been performed in multiple studies using different patient populations and slightly different methods. Miller, in the original description of $FEV_1$-Q, concluded $FEV_1$-Q was superior to alternative indices due to higher hazard ratio for mortality in a heterogenous clinical cohort in the UK, however there were overlapping confidence intervals as in our study. Similar studies in patients with COPD and the elderly had similar results [3,6]. A recent study focused on this topic using NHANES III and United Network for Organ Sharing (UNOS) databases concluded $FEV_1$-Q was superior to alternative methods based on higher c-statistic [16]. The accumulating

collective results indicate that $FEV_1$ indices that do not directly rely on reference ranges have equal or greater association with all-cause mortality.

A related question that our results help to shed light on is whether indices that result from transformations of raw $FEV_1$ are superior to $FEV_1$ alone, and what physiologic or statistical reason explains that difference. In our unadjusted results $FEV_1$-Q, $FEV_1/Ht^2$, and $FEV_1/Ht^3$ had higher magnitude HR and higher c-statistics, but those findings did not hold after adjustment. A commonality between $FEV_1$-Q, $FEV_1/ht^2$, and $FEV_1/Ht3$ is that they are derived by dividing $FEV_1$ by different values. The "Q", or quotient, in $FEV_1$-Q was originally conceptualized as the lower limit of survivable $FEV_1$, defined as the 1st percentile of $FEV_1$ in that population. However, a recent study showed that the 1st percentile of predicted $FEV_1$ varies by population, being about 1L in NHANES III and 0.3L in the UNOS database [16]. That study, using 0.5L and 0.4L for men and women respectively instead of their derived 1st percentiles, still showed strong association with mortality [16]. This raises a question of whether improvements in predictive power of $FEV_1$-Q over raw $FEV_1$ is reliant on a physiologic parameter or whether it is a statistical result of dividing $FEV_1$ by a scalar. This line of questioning extends to $FEV_1/Ht^2$ and $FEV_1/Ht^3$, whether any predictive benefit relies on the relationship between $FEV_1$ and height itself or its statistical role as a scalar. Our study was not designed to address these questions.

While reference-range-independent indices did not clearly show benefit for predicting mortality in adjusted analyses due to relatively wide and overlapping confidence intervals, they performed as well or better than traditional indices and potentially offer other clinical benefits. The utility of using reference-range-independent indices relates in their simplicity and dependence on patient related values. These properties help to simplify interpretation and mitigate complications that arise from applying reference ranges regarding race and ethnicity, age, and gender. However, these indices have not, to our knowledge, been tested to identify and diagnose respiratory diseases nor have clinically relevant cut-points been established for these indices. The use of reference-range-dependent indices is the current standard of practice to identify respiratory impairment and contribute to disease diagnosis. More work is needed to clarify the utility of reference-range-independent indices in diagnosis of respiratory disease and patient-related outcomes across disease categories.

## Conclusions

Our study demonstrates that reference-range-independent indices of $FEV_1$, $FEV_1$-Q, $FEV_1/ht^2$, and $FEV_1/ht^3$ had an equivalent or stronger association with all-cause mortality compared with reference-range-dependent indices $FEV_1$-Z and $FEV_1$-PP in a healthy U.S. cohort. These results were consistent in subgroup analyses of participants with airflow obstruction. Future directions include comparing $FEV_1$ indices in diagnostic utility and association with other respiratory outcomes.

## Author contributions

**Conceptualization:** Stephen Russell, Mohleen Kang, Jordan A. Kempker.

**Data curation:** Jordan A. Kempker.

**Formal analysis:** Jordan A. Kempker.

**Investigation:** Stephen Russell, Mohleen Kang, Jordan A. Kempker.

**Methodology:** Stephen Russell, Mohleen Kang, Jordan A. Kempker.

**Supervision:** Jordan A. Kempker.

**Writing – original draft:** Stephen Russell, Jordan A. Kempker.

**Writing – review & editing:** Stephen Russell, Mohleen Kang, Jordan A. Kempker.

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
