## [Decision Letter · Decision Letter 0]

28 Oct 2024

PONE-D-24-19654An Updated Comparison of Standard and Novel FEV1 Indices’ Association with All-Cause MortalityPLOS ONE

Dear Dr. Russell,

Thank you for submitting your manuscript to PLOS ONE. After careful consideration, we feel that it has merit but does not fully meet PLOS ONE’s publication criteria as it currently stands. Therefore, we invite you to submit a revised version of the manuscript that addresses the points raised during the review process.

 Apologies for a late decision as we were having difficulties to find reviewers during the summer. Please see the reviewers' comments below and respond.

We look forward to receiving your revised manuscript.

Kind regards,

Anindita Dutta, Ph.D.

Academic Editor

PLOS ONE

Journal Requirements:

“STR received funding from NHLBI T32 grant 5T32HL116271. MK received funding from the Department of Veterans Affairs VISN 7 Research Development Award.”

3. Please include captions for your Supporting Information files at the end of your manuscript, and update any in-text citations to match accordingly. Please see our Supporting Information guidelines for more information: http://journals.plos.org/plosone/s/supporting-information .

Reviewers' comments:

Reviewer's Responses to Questions

**Comments to the Author**

1. Is the manuscript technically sound, and do the data support the conclusions?

Reviewer #1: Yes

Reviewer #2: Yes

2. Has the statistical analysis been performed appropriately and rigorously? 

Reviewer #1: Yes

Reviewer #2: Yes

3. Have the authors made all data underlying the findings in their manuscript fully available?

Reviewer #1: Yes

Reviewer #2: Yes

4. Is the manuscript presented in an intelligible fashion and written in standard English?

Reviewer #1: Yes

Reviewer #2: Yes

5. Review Comments to the Author

Reviewer #1: This is a timely and important study highlighting the limitations of utilizing Z-scores in interpreting PFTs, and comparing several approaches to PFT interpretation including raw FEV1, FEV1-Z scores, FEV1/H2 and FEV1/H3. The authors utilized the NHANES database to evaluate the association of different FEV1 indices with all-cause mortality. They found that FEV1/ht3 might have the best utility predicting all-cause mortality. The strength of the study is tackling an important topic, and utilizing large database and rigorous statistical methods to answer whether either one of FEV1 indices is more predictive of mortality.

However, the result section is somewhat difficult to read and interpret, and might benefit from simplification and additional explanation why did authors decide to divide FEV1 indices into deciles to evaluate association with mortality. Additionally, the authors should elaborate more on why they believe that their results were significant in unadjusted analysis but were not conclusive when adjusted for sex, age and tobacco use, and whether they believe the FEV1 indices therefore offer any information beyond these clinical indices. Additionally, the authors should comment on differences between looking at all NHANES cohort and then on patients with airflow limitations only.

The authors correctly identify a major weakness of the study which is only looking at mortality and not other clinically relevant outcomes, and should consider looking at frequency of exacerbations and hospitalizations either in NHANES or in a smaller cohort of patients with airflow obstruction. Since NHANES database includes lot of patients without pulmonary problems, the mortality might confounded by a lot of other factors and not directly related to effects of airflow limitation.

The authors should also elaborate on where they see clinical utility of FEV1/ht3 compared to Z-score, FEV1 or other indices of severity. Since their study focuses on mortality and not diagnostics or management, do they expect to use FEV1/h3 to better identify high risk patients and possibly change clinical management based on that?

Overall, this study raises an important topic, however the authors should focus more on potential clinical utility of their findings.

Reviewer #2: GENERAL COMMENTS

This a well-written manuscript reporting the results of a careful analysis of the associations between various FEV1 indices and all-cause mortality in a large NHANES population. The finding that non-traditional reference-range-independent indices performed as well or better than the traditionally employed reference-range-dependent indices is a potentially impactful contribution to the interpretation of spirometry results. The authors do a nice job of discussing the potential contextual limitation of their findings -- that all-cause mortality may not be the best outcome to study if one is interested in more respiratory specific outcomes re: severity and impairment rating.

Although the authors state that the inclusion of only spirometry which meets ATS A and B grades is a strength in terms of reliability, it is also a limitation in terms of generalizability. It is well known that persons who have respiratory symptoms perform spirometry less reliably than asymptomatic persons. Including grade C spirometry results in a sensitivity analysis would address this limitation.

SPECIFIC COMMENTS

lines 106-107 The sentence is unclear to which category of indices that "Such indexing strategies..." refers, reference-range-dependent or reference-range-independent. In addition, one reference-range-independent index, FEV1-Q, is also derived from large population samples.

samples of individuals which are costly to maintain.

line 140 The name, model number, and manufacturer of the spirometer used should be given.

6. PLOS authors have the option to publish the peer review history of their article (what does this mean? ). If published, this will include your full peer review and any attached files.

**Do you want your identity to be public for this peer review?** For information about this choice, including consent withdrawal, please see our Privacy Policy .

Reviewer #1: No

Reviewer #2: No

---

## [Author Response · Author response to Decision Letter 1]

7 Mar 2025

An Updated Comparison of Standard and Novel FEV1 Indices’ Association with All-Cause Mortality: Response to Review. Our comments are bolded and italicized.

Reviewer #1: This is a timely and important study highlighting the limitations of utilizing Z-scores in interpreting PFTs, and comparing several approaches to PFT interpretation including raw FEV1, FEV1-Z scores, FEV1/H2 and FEV1/H3. The authors utilized the NHANES database to evaluate the association of different FEV1 indices with all-cause mortality. They found that FEV1/ht3 might have the best utility predicting all-cause mortality. The strength of the study is tackling an important topic, and utilizing large database and rigorous statistical methods to answer whether either one of FEV1 indices is more predictive of mortality.

Specific Reviewer Comments:

1. However, the result section is somewhat difficult to read and interpret, and might benefit from simplification and additional explanation why did authors decide to divide FEV1 indices into deciles to evaluate association with mortality.

We agree with the reviewer and strove for greater simplicity in the results. We did several things to improve readability. We removed p-values and c-statistics since all p-values presented were significant and also available in the provided tables. We removed c-statistics because they are very similar across FEV1 indices in the primary analyses, available in the tables, and discussed in the text. Finally, we consolidated the subgroup and sensitivity analyses and streamlined their presentation by removing the specific hazard ratios and confidence intervals. These are available in the supplementary tables and did not substantially change the overall study interpretations. For consistency the methods sections was also changed to combine the subgroup and sensitivity analyses.

Regarding the comment about FEV1 deciles, these were grouped to allow direct comparison between indices. The indices’ differing units, scales, and cutpoints would otherwise make comparisons difficult. Furthermore, choosing 10 groupings (rather than a smaller number such as quartiles or quintiles) provides a greater spread to potentially view inflection points, thus providing more granularity. This is similar to the methodology used by Miller in the original FEV1Q article and others that have approached this topic. I have added this sentence to the Methods, lines 168-170, “Categorizing each index into deciles is a methodology used in previous papers to compare multiple indices that may have different or unknown severity cutpoints”.

2. Additionally, the authors should elaborate more on why they believe that their results were significant in unadjusted analysis but were not conclusive when adjusted for sex, age and tobacco use, and whether they believe the FEV1 indices therefore offer any information beyond these clinical indices.

The final paragraph of the discussion addresses this issue, which we agree is a key clinical question. Our data add to the discussion but are not comprehensive enough to determine whether reference-range-independent indices offer additional information beyond traditional indices across different pulmonary diseases and clinical outcome apart from mortality.

I have added the following sentence to the lines 404-406, the first sentences of the last paragraph, “While reference-range-independent indices did not clearly show benefit for predicting mortality in adjusted analyses due to relatively wide and overlapping confidence intervals, they performed as well or better than traditional indices and potentially offer other clinical benefits.”

3. Additionally, the authors should comment on differences between looking at all NHANES cohort and then on patients with airflow limitations only.

We agree and have added the following sentence to the discussion, “In subgroup analysis of subjects with airflow limitation results were similar, however caution is warranted due to very wide confidence intervals.” Due to the very wide confidence intervals, as well as abundance of other cohorts (SPIROMICS, COPDgene, etc) with larger populations of patients with airflow limitation, this question is best explored in those cohorts.

4. The authors correctly identify a major weakness of the study which is only looking at mortality and not other clinically relevant outcomes, and should consider looking at frequency of exacerbations and hospitalizations either in NHANES or in a smaller cohort of patients with airflow obstruction.

We agree with this comment and it is an area for future study, however we have not made specific changes to the text in response to this comment. We feel this would be best studied in other COPD cohorts where there was intentional recruitment of subjects with airflow obstruction and collection of additional important patient-related outcomes such as symptoms, functional status, and exacerbations.

5. Since NHANES database includes lot of patients without pulmonary problems, the mortality might confounded by a lot of other factors and not directly related to effects of airflow limitation.

We agree with this comment but not have made specific changes in the text. Most likely the bulk of deaths were non-pulmonary, and as noted by this reviewer NHANES is a nationally representative, relatively healthy cohort. We feel that our analysis is still useful since FEV1 is known to be associated with mortality beyond patients with specific pulmonary disease. FEV1 is not likely to be causally related to mortality in these patients, but still offers clinical utility due to the strength of association with mortality. Additionally, currently accepted FEV1 indices such as z-scores cutpoints are determined by examining distributions of cohorts reportedly free from pulmonary diseases without relationship to symptoms or outcomes and thus we also think that the line of inquiry of our and others’ work in trying to examine these indices to outcomes remains an important one. We certainly agree that examining these issues with other pulmonary-specific outcomes and in both disease free and those with known pulmonary conditions all have merit for such an ubiquitous test.

6. The authors should also elaborate on where they see clinical utility of FEV1/ht3 compared to Z-score, FEV1 or other indices of severity. Since their study focuses on mortality and not diagnostics or management, do they expect to use FEV1/h3 to better identify high risk patients and possibly change clinical management based on that?

Overall, this study raises an important topic, however the authors should focus more on potential clinical utility of their findings.

We appreciate this comment as all of the authors are clinicians and are interested in scientific work that will ultimately have a positive effect on the assessment and management of patients. However, for many of the reasons addressed in points above we think it is too soon to draw definitive conclusions about the clinical implications of the comparative utility of these indices. There is still much work to do to assess their relative utility in predicting symptoms, functional status, and exacerbations (as reviewer one pointed out) in both cross-sectional and longitudinal analyses and among healthy and specific pulmonary patient populations. We see our work as part of a body and trajectory of work that along with others’, will help to answer some of these questions and be able to make more definitive statements once the data is out.

Reviewer #2: GENERAL COMMENTS

This a well-written manuscript reporting the results of a careful analysis of the associations between various FEV1 indices and all-cause mortality in a large NHANES population. The finding that non-traditional reference-range-independent indices performed as well or better than the traditionally employed reference-range-dependent indices is a potentially impactful contribution to the interpretation of spirometry results. The authors do a nice job of discussing the potential contextual limitation of their findings -- that all-cause mortality may not be the best outcome to study if one is interested in more respiratory specific outcomes re: severity and impairment rating.

1. Although the authors state that the inclusion of only spirometry which meets ATS A and B grades is a strength in terms of reliability, it is also a limitation in terms of generalizability. It is well known that persons who have respiratory symptoms perform spirometry less reliably than asymptomatic persons. Including grade C spirometry results in a sensitivity analysis would address this limitation.

We agree with this comment and have performed an additional sensitivity analysis with Grade C and D added. The analysis can be found in supplement tables E7 and E8. This additional analysis requires explanation principally in the methods and results sections, with details below:

Within the methods, the following sentence was edited at the end of the Spirometry section, lines 146-147, “Participants meeting grades A and B were included in our primary analysis, and subjects meeting grades A, B, C, and D were included in a sensitivity analysis.” In the subgroup and sensitivity analysis subheading of the methods section, lines 196-198 we have included the sentences “Second, we included subjects with FEV1 ATS grades A, B, C, and D. Our primary analysis included only ATS quality grade A and B to increase internal validity, however grades lower than B are encountered in clinical practice.” Within the results section, lines 272-274 we added “A second sensitivity analysis included 1,163 additional participants with lower quality spirometry grades C and D. This analysis of the 14,157 participants with ATS grades A, B, C, and D showed similar results the primary analysis (see Supplement Tables E7 and E8 for full results).”

SPECIFIC COMMENTS

2. lines 106-107 The sentence is unclear to which category of indices that "Such indexing strategies..." refers, reference-range-dependent or reference-range-independent.

We agree with this comment and have edited that sentence as follows, which is now found on lines 109-110:

The sentence read: “Such indexing strategies for spirometry rely on derived values or equations from large samples of individuals which are costly to maintain and carry the potential of bias when applied to other external populations”

In has been changed to: “Development and maintenance of reference ranges for spirometry rely on large samples of individuals which are costly and carry the potential of bias when applied to other, external populations”

3. In addition, one reference-range-independent index, FEV1-Q, is also derived from large population samples. samples of individuals which are costly to maintain.

We appreciate this thoughtful comment as we discussed how to approach this issue at length before the original draft of this manuscript. On one hand, the Q was determined in an observational study on combined large population samples and thus there is a hypothesis that it is referent-populations. On the other hand, our understanding of the conceptual framework of the Q is that it was developed so as to represent a physiologic constant, below which pulmonary function and survival were thought to be limited and thus represented a physiologic nadir to be indexed to rather than a population reference value to be constantly re-evaluated in large cohorts. While we tend to agree with you, at this time no changes in the text have been made in reference to this comment given these differing opinions on this subject and it being a minor discussion point in our manuscript.

4. line 140 The name, model number, and manufacturer of the spirometer used should be given.

The sentence in question on lines 145-146 was changed to: “NHANES used an Ohio Sensormed 827 dry rolling seal spirometer through all the selected survey years and testing was completed according to the ATS standards for acceptability and reproducibility at the time of study enrollment.”

Additionally, reference #13 was added. This identifies the NHANES spirometer.

---

## [Decision Letter · Decision Letter 1]

13 Apr 2025

An Updated Comparison of Standard and Novel FEV1 Indices’ Association with All-Cause Mortality

PONE-D-24-19654R1

Dear Dr. Russell,

We’re pleased to inform you that your manuscript has been judged scientifically suitable for publication and will be formally accepted for publication once it meets all outstanding technical requirements.

Please add/correct the following in your manuscript before submission:

1. Why were the grade D results from the sensitivity analysis included?

2. line 163 Should be "each FEV1 INDEX decile group"

3. line 274 Should be "intrinsic abnormalities OF lung function"

Kind regards,

Anindita Dutta, Ph.D.

Academic Editor

PLOS ONE

Additional Editor Comments (optional):

Reviewers' comments:

Reviewer's Responses to Questions

**Comments to the Author**

1. If the authors have adequately addressed your comments raised in a previous round of review and you feel that this manuscript is now acceptable for publication, you may indicate that here to bypass the “Comments to the Author” section, enter your conflict of interest statement in the “Confidential to Editor” section, and submit your "Accept" recommendation.

Reviewer #1: All comments have been addressed

Reviewer #2: (No Response)

2. Is the manuscript technically sound, and do the data support the conclusions?

Reviewer #1: Yes

Reviewer #2: Yes

3. Has the statistical analysis been performed appropriately and rigorously? 

Reviewer #1: Yes

Reviewer #2: Yes

4. Have the authors made all data underlying the findings in their manuscript fully available?

Reviewer #1: Yes

Reviewer #2: Yes

5. Is the manuscript presented in an intelligible fashion and written in standard English?

Reviewer #1: Yes

Reviewer #2: Yes

6. Review Comments to the Author

Reviewer #1: The authors adequately addressed some concerns of the reviewers by adding comments, edits and sub-group analysis results. There is still question of future utility of this specific index however as authors state this might be better addressed in future studies looking at specific events such as exacerbations, or specific patient subpopulations.

Reviewer #2: GENERAL COMMENTS

The authors have done a nice job in responding to the Reviewer comments. I appreciate the addition of the sensitivity analysis with less well performed spirometry. My only remaining issue is why were grade D spirometry results used. Grade C results make sense to me, but not grade. D results. The authors should either drop the grade D results from the sensitivity analysis or explain why they were included.

SPECIFIC COMMENTS

line 163 Should be "each FEV1 INDEX decile group"

line 274 Should be "intrinsic abnormalities OF lung function"

7. PLOS authors have the option to publish the peer review history of their article (what does this mean? ). If published, this will include your full peer review and any attached files.

**Do you want your identity to be public for this peer review?** For information about this choice, including consent withdrawal, please see our Privacy Policy .

Reviewer #1: **Yes: ** Eva Otoupalova

Reviewer #2: No

---

## [Editor Report · Acceptance letter]

PONE-D-24-19654R1

PLOS ONE

Dear Dr. Russell,

I'm pleased to inform you that your manuscript has been deemed suitable for publication in PLOS ONE. Congratulations! Your manuscript is now being handed over to our production team.

Kind regards,

on behalf of

Dr. Anindita Dutta

Academic Editor

PLOS ONE